# 12q21 Interstitial Deletions: Seven New Syndromic Cases Detected by Array-CGH and Review of the Literature

**DOI:** 10.3390/genes13050780

**Published:** 2022-04-27

**Authors:** Maria Paola Recalcati, Ilaria Catusi, Maria Garzo, Serena Redaelli, Marta Massimello, Silvia Beatrice Maitz, Mattia Gentile, Emanuela Ponzi, Paola Orsini, Anna Zilio, Annamaria Montaldi, Annapaola Calò, Anna Paola Capra, Silvana Briuglia, Maria Angela La Rosa, Lucia Grillo, Corrado Romano, Sebastiano Bianca, Michela Malacarne, Martina Busè, Maria Piccione, Lidia Larizza

**Affiliations:** 1Laboratorio di Citogenetica e Genetica Molecolare, IRCCS Istituto Auxologico Italiano, 20095 Cusano Milanino, Italy; ilaria.catusi@gmail.com (I.C.); maria.garzo@libero.it (M.G.); l.larizza@auxologico.it (L.L.); 2Dipartimento di Chirurgia e Medicina Traslazionale, Università di Milano-Bicocca, 20126 Milan, Italy; serena.redaelli@unimib.it; 3Ospedale San Gerardo, Unità di Genetica Pediatrica, Fondazione Monza e Brianza per il Bambino e la Sua Mamma (MBBM), 20900 Monza, Italy; marta.massimello@unimi.it (M.M.); maitz.silvia@gmail.com (S.B.M.); 4UOC Genetica Medica, PO Di Venere, ASL Bari, 70131 Bari, Italy; mattia.gentile@asl.bari.it (M.G.); emanuela.ponzi@asl.bari.it (E.P.); paola.orsini@asl.bari.it (P.O.); 5U.O.S. Laboratorio di Genetica, Azienda ULSS 8 Berica, 36100 Vicenza, Italy; anna.zilio@aulss8.veneto.it (A.Z.); anna.montaldi@aulss8.veneto.it (A.M.); annapaola.calo@aulss8.veneto.it (A.C.); 6Dipartimento di Scienze Biomediche, Odontoiatriche e Delle Immagini Morfologiche e Funzionali, Università di Messina, 98125 Messina, Italy; acapra@unime.it (A.P.C.); silvana.briuglia@unime.it (S.B.); maria.larosa@unime.it (M.A.L.R.); 7Research Unit of Rare Diseases and Neurodevelopmental Disorders, Oasi Research Institute-IRCCS, 94018 Troina, Italy; lgrillo@oasi.en.it (L.G.); cromano@oasi.en.it (C.R.); 8Medical Genetics, Section of Medical Biochemistry, Department of Biomedical and Biotechnological Sciences, University of Catania, 95124 Catania, Italy; 9Genetica Medica, ARNAS Garibaldi, 95123 Catania, Italy; sebastianobianca@bgenetica.it; 10IRCCS Istituto Giannina Gaslini, UOC Laboratorio di Genetica Umana, 16147 Genova, Italy; michelamalacarne@gaslini.org; 11U.O.C. Genetica Medica AOOR Villa Sofia-Cervello, 90146 Palermo, Italy; martina.buse87@gmail.com; 12Dipartimento di Scienze della Promozione della Salute, Materno-Infantile, Medicina Interna e Specialistica d’Eccellenza, “G. D’Alessandro” (PROMISE), Università Degli Studi di Palermo, 90144 Palermo, Italy; maria.piccione@unipa.it

**Keywords:** array-CGH, 12q21 deletion, copy number variants (CNVs), variation intolerant genes, loss of function, developmental delay/intellectual disability (DD/ID), congenital anomalies, dysmorphisms, genetic counseling, patient management

## Abstract

Interstitial deletions of the long arm of chromosome 12 are rare, with a dozen patients carrying a deletion in 12q21 being reported. Recently a critical region (CR) has been delimited and could be responsible for the more commonly described clinical features, such as developmental delay/intellectual disability, congenital genitourinary and brain malformations. Other, less frequent, clinical signs do not seem to be correlated to the proposed CR. We present seven new patients harboring non-recurrent deletions ranging from 1 to 18.5 Mb differentially scattered across 12q21. Alongside more common clinical signs, some patients have rarer features such as heart defects, hearing loss, hypotonia and dysmorphisms. The correlation of haploinsufficiency of genes outside the CR to specific signs contributes to our knowledge of the effect of the deletion of this gene-poor region of chromosome 12q. This work underlines the still important role of copy number variations in the diagnostic setting of syndromic patients and the positive reflection on management and family genetic counseling.

## 1. Introduction

Interstitial deletions in 12q21 have been described in a dozen patients who were all carriers of non-recurrent copy number variants (CNVs) [1,2,3,4,5,6,7,8,9,10]. Despite common features, such as developmental delay and/or intellectual disability (DD/ID), congenital genitourinary, brain malformations and ectodermal anomalies, the clinical presentation is characterized by wide clinical expressivity with severity and signs depending on the size and location of the deletions and on the involved genes. In most cases the deletions are large, making it difficult to assign the causative genes to individual phenotypic signs.

A critical region (CR) in 12q21.2 has recently been narrowed down to 1.6 Mb including the four genes Synaptogamin 1 (*SYT1*, OMIM *185605), PRKC Apoptosis WT1 Regulator (*PAWR*, OMIM *601936), Protein Phosphatase 1 Regulatory Subunit 12A (*PPP1R12A*, OMIM *602021) and Otogelin-like protein (*OTOGL*, OMIM *614925). Of these four, the major candidates seem to be *SYT1*, encoding a highly conserved synaptic vesicle protein, and *PPP1R12A*, encoding a regulator of myosin phosphatase acting in neurotransmitter release [5].

A distal critical region in 12q21.33, encompassing the four Decorin (*DCN*, OMIM *125255), Keratocan (*KERA*, OMIM *603288), Lumican (*LUM*, OMIM *600616) and Epiphycan (*EPYC*, OMIM *601657) genes, defines the contiguous gene deletion syndrome PACD (posterior amorphous corneal dystrophy, OMIM #612868), for which *DCN* appears as the most relevant gene according to haploinsufficiency prediction (Appendix A) [10].

In this study we describe seven new patients with deletions detected by array-based comparative genomic hybridization (array-CGH), collected through a cooperation organized by the Cytogenetics and Cytogenomics Working Group of the Italian Society of Human Genetics (SIGU). The involved regions spanning from 1 to 18.53 Mb are differentially positioned in the 12q21.1–q21.33 region. Three deletions overlap the 1.6 Mb CR in 12q21.2, confirming its potential causality for the aforementioned clinical features. The other four deletions, relatively smaller (1 Mb to 6.75 Mb), map distally, hence providing the tool to investigate the contribution of genes outside the CR on the clinical phenotype. Indeed, the carrier patients manifest rare but recurrent clinical signs such as heart defects, hearing loss, hypotonia and facial dysmorphisms including down-slanting palpebral fissures, wide mouth and prominent forehead.

The genotype–phenotype correlations in our patients, compared with cases in the literature and in the DECIPHER database, bring to light relationships between the deletion of specific 12q21 genes and distinct phenotypic traits.

## 2. Materials and Methods

The seven investigated patients underwent molecular cytogenetics analyses as routine diagnostic procedures in six different Italian cytogenetics laboratories. In all cases, informed consent signed by the patients’ parents was provided. DNAs were isolated from peripheral blood samples. Array-CGH analyses were performed using: Agilent SurePrint G3 Oligo ISCA v2.0 4 × 180 K, resolution ~25 kb (Patients 1 and 3); Technogenetics CytoChip ISCA 8 × 60 K v2.0, resolution ~100 kb (Patient 2); Agilent SurePrint G3 Human CGH Microarray kit 8 × 60 K, resolution ~100–150 kb (Patients 4 and 5); CytoSure Oligo array ISCA v2 4 × 180 K OGT (Oxford Gene Technology), resolution ~80 kb (Patient 6); and Bluegnome 4 × 180 K CytoChip Oligo ISCA, resolution ~75–100 kb (Patient 7). Protocols provided by the suppliers have been followed without modification. Nucleotide designations were assigned according to the GRCh37/hg19 assembly of the human genome. BAC—FISH (bacterial artificial chromosome—fluorescence in situ hybridization) experiments for Patient 2 and Patient 7, with RP11-148D15 and RP11-76B1 clones, respectively, were performed according to Lichter et al. (1988) with minor modifications [11]. For Patient 5, whole exome sequencing (WES) analysis was performed on genomic DNA isolated by peripheral blood of the trio (index patient and parents), enriched for the exonic regions (SureSelect XT2 Clinical Research Exome, Agilent Technologies, Santa Clara, CA, USA), followed by sequencing with 150-bp paired-end reads on the NextSeq500 platform (Illumina, San Diego, CA, USA).

The 12q21 region gene content has been assessed merging the bioinformatics data from UCSC Genome Browser, DECIPHER, OMIM and gnomAD databases [12,13,14,15].

## 3. Results

### 3.1. Array-CGH

Our study includes seven patients, five males and two females, with age at diagnosis ranging from 4 to 16 years. Array-CGH was performed on all seven patients and the results are provided in Figure 1 (red bars) and Figure 2, while the clinical features of each different 12q microdeleted patient are detailed in Table 1. Patients’ 2 and 7 deletions have been confirmed by BAC—FISH experiments and Figure 2 shows the images for Patient 2. Patients 2 and 3 have been extrapolated from a cohort of 5110 Italian patients, collected by the SIGU Cytogenetics and Cytogenomics Working Group and analyzed by array-CGH as reported by Catusi et al. (2020) [16].

The inheritance pattern could be determined in four of the seven patients: the microdeletion arose de novo in Patients 5, 6 and 7 and was transmitted to Patient 2 by her mother, also with mild intellectual disability and minor dysmorphisms.

The extent of microdeletions ranges from the smallest 1 Mb deletion of Patient 7 in 12q21.32q21.33, only containing five coding genes, to the largest 18.5 Mb deletion of Patient 1 in 12q21.1q21.33, comprising 46 coding genes (Figure 1 and Table 1).

None of the deleted regions overlap with established benign losses reported in the Database of Genomic Variants (DGV) [17].

### 3.2. 12q21 Region and Synopsis of Relevant Genes

The 12q21 region spans 21.2 Mb and includes 60 coding genes with a gene density remarkably lower relative to the entire chromosome 12 (2.8 vs. 7.5 coding genes/Mb, respectively). Such a gene density is below even that of chromosome 21, which is considered the poorest gene-containing chromosome (4.5 coding genes/Mb) [13]. To predict the deleterious effect of a deletion, the metrics of haploinsufficiency (HI) and loss-of-function intolerance (pLI) were examined, and 20 of the 60 coding genes in 12q21 stood out as most interesting. Genes with HI score ≤ 10% and/or pLI score ≥ 0.9 are listed in Appendix A that also includes disease-causing genes matching the clinical features of our and literature patients.

### 3.3. Literature and DECIPHER Patients

Twelve patients are reported in the literature with deletions characterized by array-CGH analysis [1,2,3,4,5,6,7,8,9,10]. Ten deletions (ranging from 5.5 to 25.3 Mb) partially overlap in the 12q21 region and have recently permitted to delimit a 1.6 Mb CR comprising four genes, among which *SYT1* and *PPP1R12A* are defined as major candidates [5]. Regarding the two remaining deletions, the one described by Akilapa et al. in 2015 lies in 12q21.31q21.32 region and does not overlap with the CR [9], whereas the smallest 1.3 Mb deletion, including the four *DCN*, *KERA*, *LUM* and *EPYC* genes, defines the contiguous gene deletion syndrome PACD (posterior amorphous corneal dystrophy; OMIM 612868) [10].

A number of 12q21 deletions are reported in the DECIPHER database. We have extrapolated the most informative ones according to patients’ phenotype and the region involved. The smallest deletion, sized 257 kb, only involves *PPP1R12A* (Patient 291358), while the largest spans 11 Mb and includes 50 coding genes (Patient 1875) [13].

Figure 1 reports the deletions known from the literature (black bars), the DECIPHER deletions (purple bars), and the seven deletions reported in the present study (red bars). Patients’ clinical features are shown in Table 2 for literature patients and in Table 3 for DECIPHER patients.

## 4. Discussion

We surveyed seven new patients carrying microdeletions sized from 1 Mb to 18.53 Mb, differentially positioned in the 12q21.1–q21.33 region. All patients have DD/ID, which is confirmed as the most common clinical sign. In three patients (1, 2 and 3) the microdeletions partially overlap the 1.6 Mb CR recently proposed, and support the recurrent clinical signs, namely, DD/ID, growth retardation, brain and genitourinary defects and ectodermal anomalies, associated with 12q21 deletions [5]. The CR is currently the minimal overlapping region among literature patients. Since the reported deletions are mainly wide, spanning from 5.5 to 25.3 Mb, the association of the deletion of specific 12q21 genes to distinct phenotypic traits is still a challenge.

As mentioned above, the proposed CR encompasses four genes: *SYT1*, *PAWR*, *PPP1R12A* and *OTOGL*, with *SYT1* and *PPP1R12A* as the most convincing candidates [5].

*SYT1* encodes for Synaptotagmin 1, a protein highly and exclusively expressed in the brain and involved in trafficking of synaptic vesicles at synapses. A *SYT1* HI score of 3.9% is far below the upper threshold (10%) fixed for haploinsufficiency [13,15]. Heterozygous *SYT1* mutations are associated with Backer–Gordon syndrome (BAGOS; OMIM #618218) which is characterized by DD, motor impairment and dysmorphisms such as high forehead, smooth philtrum and short nose [18]. The same features recur in 12q21 deletion patients including the *SYT1* gene. The only patient with a small deletion affecting the sole *SYT1* gene is DECIPHER Patient 331592, who shows only DD/ID (Figure 2 and Table 3).

*PPP1R12A*, a developmental gene encoding the regulatory subunit 12 of phosphatase, involved in cell migration, adhesion, and morphogenesis, has a HI score of 2.14% and a pLI score of 1, hence being placed at the top of the dosage sensitive genes within the CR. *PPP1R12A* loss-of-function heterozygous pathogenic variants have been associated with a congenital malformation syndrome affecting the embryogenesis of genitourinary and brain systems (GUBS; OMIM *618820) including sex development disorders. Carriers also show dysmorphisms such as micro- or macrocephaly, upslanting palpebral fissures, long philtrum, micrognathia, low-set ears, hypertelorism, strabismus and small nose [19]. No sex development disorders have been recorded for 12q21 deletions involving *PPP1R12A*, but brain and genitourinary defects are recurrent. Among our patients, Patient 2 has horseshoe kidney and several genitourinary defects, as already reported in literature patients [2,5,7]. Moreover, the same dysmorphisms are present in all reported patients whose deletions include *PPP1R12A*, and in DECIPHER Patients 1582 and 291358. This latter patient carries a 257 kb deletion involving *PPP1R12A* exons from 2 to 25.

The deletions of our Patients 1, 2 and 3 confirm the CR by Niclass et al. [5], including the two major candidate genes. Furthermore, clinical data are consistent with a concurrent effect of *SYT1* and *PPP1R12A* deletions, which are likely to be mainly responsible for the neurodevelopmental issues and for the congenital defects and dysmorphism, respectively. Further single gene deletions and functional studies could help to assess the sensitivity of different tissues to the loss of either gene, and to further investigate their role in clinical signs etiology.

However, Patients 4, 5, 6 and 7 manifest DD/ID, although their deletions do not overlap with the CR region. It could be that other genes included in these deletions could be implicated in DD/ID onset, such as *TMTC2*, *SLC6A15*, *CEP290*, *TMTC3* and *NTS*, which are highly expressed in the brain [12] and predicted as dosage-sensitive by HI or pLI scores (Appendix A).

Conversely, on the basis of current knowledge, *SYT1* and *PPP1R12A* do not seem related to the ectodermal anomalies observed in 12q21 deleted patients and, moreover, none of the genes identified appear to be specifically expressed in the skin. Considering the deletions of patients showing ectodermal anomalies, we hypothesize that the causative gene(s) for skin features resides within a region of about 4.7 Mb comprised between the start of the deletion of our Patient 3 and the end of the deletion of Patient 3 of Niclass et al. [5] (nucleotides 78634011-83378230; Figure 2). Further studies and patients are needed to corroborate this hypothesis.

Patients 4 and 6 also present interatrial defects (Table 1) consistent with some literature patients [2,4,7,8] and DECIPHER Patient 261816. Overlap of these patients’ deletions point to *TMTC2* and *SLC6A15* as candidate genes, though no expression or mutation data are yet available to associate these genes to cardiac defects.

Our Patients 5 and 6 carrying deletions involving *NTS* and *SLC6A15* manifest DD/ID associated with facial dysmorphisms. Patient 5 has a broad nasal base, low-hanging columella, wide mouth, down-slanting palpebral fissures, while Patient 6 has a wide mouth. Interestingly, Akilapa et al. [9] have described three members of the same family with a varying degree of ID, facial dysmorphisms as wide mouth, broad nasal base with low-hanging columella, tall stature and obesity, segregating with a 2.2 Mb deletion in 12q21.31q21.32. The Authors proposed *NTS* and *SLC6A15* as candidate genes for ID, since both are expressed in the brain and *NTS* has a predicted HI score of 1.63% (Appendix A). It is noteworthy that *ALX1*, a gene mapping between *SLC6A15* and *NTS*, is associated with autosomal recessive frontonasal dysplasia 3 (FND3, OMIM #613456), a syndrome of variable severity, characterized by a series of facial abnormalities affecting eyes, forehead and nose [20]. The *ALX1* gene can be thus proposed as a plausible candidate for the facial dysmorphisms observed in 12q21.31q21.32 deletion carriers, even though evidence of a facial phenotype has not yet been assessed in the case of heterozygous deletions. It will be useful in the future to more precisely describe patients with overlapping deletions in order to better delineate a dysmorphic phenotype resulting from heterozygous loss of function of *ALX1* and/or *NTS*.

For Patient 5, WES was also performed, revealing an additional likely pathogenic loss-of-function homozygous variant in the *ACY1* gene localized in 3p21.2: c.1057C>T (p.Arg353Cys), inherited from both parents. ACY1 encodes for the amino acylase 1 enzyme. Its deficiency results in a low hydrolysis of a series of N-acetyl-L-amino. The clinical manifestation of *ACY1* deficiency is heterogeneous and includes muscular hypotonia, brain malformations, ID and other neurologic symptoms, while normal psychomotor development has also been noted in some individuals [21]. ACY1 deficiency could explain DD/ID and hypotonia in Patient 5 but not dysmorphisms, which instead resemble those of Patient 6 and the patient in study by Akilapa et al. The phenotype of Patient 5 could be the result of the concomitance of both genetic anomalies.

Patient 7 has been diagnosed with macrosomy (CC 58 cm, >97°; height 165 cm, >90°; hand 19.5 cm, (75–97°), ID and Asperger syndrome with left hearing loss. His deletion involves *KITLG*, a mutation causative of autosomal dominant sensorineural deafness 69 (OMIM 616697) [22]. Our Patient 7 and the patient described by Klein et al. corroborate the causative role of *KITLG* for the deafness phenotype in 12q21 deleted patients, since they are the only patients with *KITLG* deletion and deafness (Table 1 and Table 2) [2].

For completeness, it is worth mentioning the contiguous genes deletion syndrome associated with corneal defects, mapping to the distal 12q21.33 region, and including the *DCN*, *KERA*, *LUM* and *EPYC* genes [10]. None of our patients’ deletions encompassed these genes and none of them manifested corneal anomalies, in contrast to the patients described by Cano et al., Lenk et al., and DECIPHER Patients 306170 and 273686, who presented corneal dystrophy, PACD (posterior amorphous corneal dystrophy), Peters anomaly and microcornea, respectively [1,10].

## 5. Conclusions

We described seven novel patients with deletions scattered across the 12q21 region. 12q21 deletions are associated with a wide spectrum of clinical signs such as DD/ID, congenital malformations of the brain, heart and genitourinary systems. Considering the CR recently proposed as responsible for the most frequent clinical signs, we performed our study firstly to strengthen the role of the two major candidate genes, *SYT1* and *PPP1R12A*, in the CR and, secondly, to investigate the effect of genes outside the CR.

Our Patients 1, 2 and 3, and others extrapolated from the DECIPHER database, carrying deletions overlapping the CR, validate the first assumption. Our Patients 5 and 6 support the hypothesis of a region outside the CR in 12q21.33, containing the *ALX1* and the *NTS* genes, associated with dysmorphism such as wide mouth, broad nasal base and low hanging columella and ID [9].

In addition, deafness, described only for our Patient 7 and the patient from Klein et al., could be caused by haploinsufficiency of *KITLG* gene, since its mutation has been already identified in patients with sensorineural deafness [2,22].

Finally, since DD/ID is present in almost all patients with deletion in 12q21 regardless of the involvement of the CR, it is possible that other genes outside the CR contribute to the neurological phenotype. In particular, *TMTC2, SLC6A15, CEP290, TMTC3, NTS* genes, all highly expressed in the brain, could be considered as additional strong candidates [12].

In the future, investigation of novel patients with deletions in 12q21 will hopefully clarify the contribution of single genes to the complex phenotype of these patients, in order to also provide adequate counseling to patients and families in both pre- and post-natal diagnostic settings. In addition, since 12q21 microdeletions do not hamper survival and reproduction, as shown by maternal inheritance of a 5.7 Mb deletion in Patient 2, early identification of these deletions could ensure a proper intervention and supporting therapy for the patients and correct genetic counseling for the involved families and their subsequent pregnancies.

Although spreading of WES or WGS (whole genome sequencing) technologies are often crucial in identifying point mutations as the genetic cause of a disorder, sometimes proposed as first tier approaches, our work further confirms that the identification of CNVs still plays a relevant role for the diagnosis of patients with complex phenotypes and the provision of standardized psychological assessments and other medical follow-ups.

## Figures and Tables

**Figure 1 genes-13-00780-f001:**
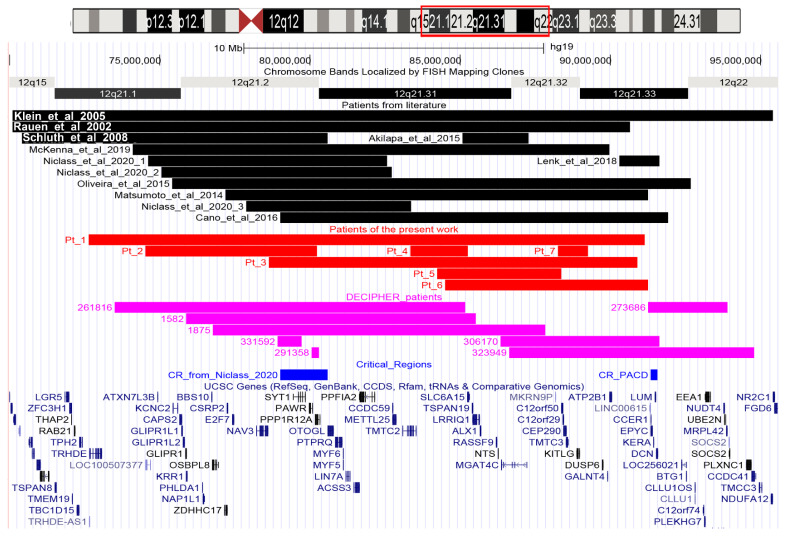
Physical map of the 12q15q22 region (nucleotides 70,000,000 to 95,600,000, GRCh37/hg19) adapted from UCSC Genome Browser: differently colored bars indicate the genomic regions involved in the microdeletions of our seven patients (red bars), deleted patients reported in the literature (black bars) and patients described in the DECIPHER database (purple bars). All coding genes included in this region are annotated. The critical regions (CR) described in the literature are indicated with blue bars. PACD = posterior amorphous corneal dystrophy [10].

**Figure 2 genes-13-00780-f002:**
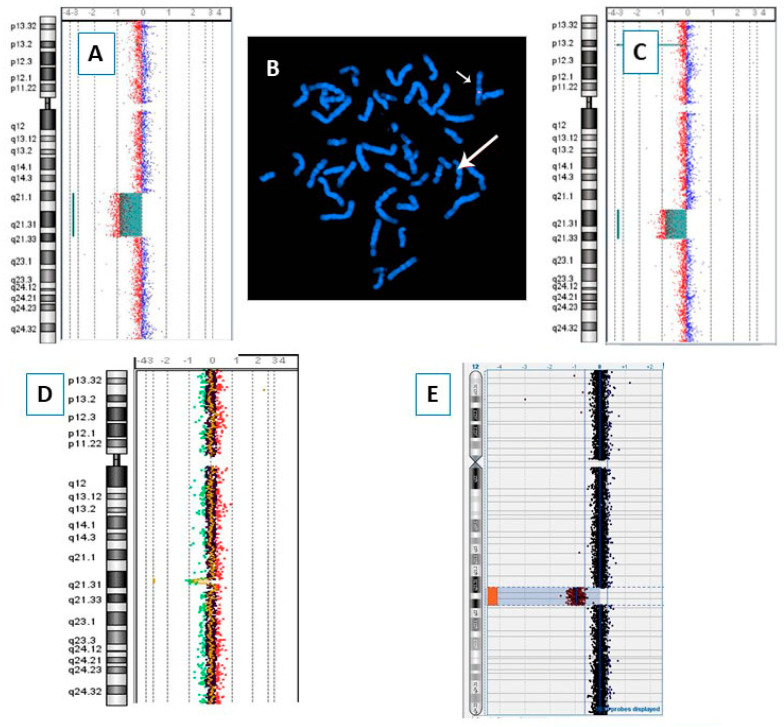
Molecular cytogenetic analyses: array-CGH profile of chromosome 12 of Patients 1 (**A**), 3 (**C**), 4 (**D**) and 6 (**E**); FISH of Patient 2 with probe RP11-148D15 (**B**); the big and the small arrows point, respectively, to the deleted and the normal chromosomes 12.

**Table 1 genes-13-00780-t001:** Clinical and molecular cytogenetics data of 12q21 deleted patients described in the present work.

Patients	1	2	3	4	5	6	7
Age at last evaluation and sex	8y (M)	11y (F)	16y (M)	11y (M)	4y (M)	7y (F)	15y (M)
Coordinates on chromosome 12 (hg19)	72634701-91163349	74536114-80234335	78634010-90918721	83359339-85260949	84234089-88380156	84507288-91264704	88264952-89267923
Deletion size (Mb)	18.53	5.7	12.28	1.9	4.15	6.76	1
Inheritance	Unknown	From mother with similar phenotype	Unknown	Unknown	de novo	de novo	de novo
Growth retardation	+	-	-	-	+	-	-
DD/ID	Learning and language impairment.	Mild ID, language impairment.	Mild DD	DD	Mild ID. Receptive and expressive language significant delayed.	Learning and language impairments.	Mild ID and language impairment.
Prominent forehead	-	-	-	-	+	+	-
Hypertelorism	-	-	+	-	-	+	-
Low set ears	-	-		-	-	+	-
Short nose	-	+	+	-	-	-	-
Other dysmorphisms	Relative macrocephaly.	Dysplastic ears, thick philtrum, large mouth.	-	-	Broad nasal base, low-hanging columella, wide mouth, down-slanting palpebral fissures. Relative microcephaly.	Upturned nose, long philtrum, ogival palate, wide mouth.	Ogival palate, eye asymmetry.
2–3 toe syndactyly + single palmar crease	-	-	-	-	-	-	-
Cardiac anomalies	-	-	-	Ostium secundum atrial septal defect.	-	Small oval fossa shunt, interatrial shunt.	-
Ectodermal abnormalities	Dry skin, sparse eyebrows.	-	+	-	-	-	-
Ocular abnormalities	Astigmatism	Exophthalmos	+	-	Epichantus	Exophoria	Astigmatism
Genitourinary anomalies	Horseshoe kidney	-	-	-	-	-	-
Brain Abnormalities	-	Anterior intrasellar arachnoid cyst.	-	-	-	-	-
Hypotonia	+	+	-	-	+	-	-
Other	Severe motor impairment.	Ligamentous laxity.	-	Non-spastic muscle contractures.	Oppositional behaviors, hetero-aggressive attitudes with refusal of body contact.		Asperger, hearing loss, macrosomia.

Y: years; M: male; F: female; “+”: present; “-”: absent; DD: developmental delay; ID: intellectual disability.

**Table 2 genes-13-00780-t002:** Clinical data of 12q21 deleted patients described in the literature.

	Klein et al., 2005 [2]	Rauen et al., 2002 [7]	Schluth et al., 2008 [8]	McKenna et al., 2019 [4]	Niclass et al., 2020_1 [5]	Niclass et al., 2020_2 [5]	Oliveira et al., 2015 [6]	Matsumoto et al., 2014 [3]	Niclass et al., 2020_3 [5]	Cano et al., 2016 [1]	Akilapa et al., 2015 [9]	Lenk et al., 2018 [10]
Growth retardation			+		+					+		
DD/ID	+	+	+	+	+		+	+	+	+	+	+
Prominent forehead	+	+	+	+	+	+	+	+				
Hypertelorism	+		+				+		+			
Low set ears	+	+	+	+	+		+	+	+			
Short nose	+	+		+	+		+	+	+			
Other dysmorphisms			Anteverted nostrils, long philtrum, thin lips, microgna-thia	Anteverted nares	Long high palpebral fissures, hypoplastic nostrils	Flat face, hypoplastic nostrils, dysplastic left ear		Long philtrum, high arched palate	Upslanted palpebral fissures, anteverted nares, wide philtrum, thin upper lip, prominent chin	Microphthalmia, microdontia	Wide nasal base, low hanging columella, thin upper lip, wide mouth, down-slanting palpebral fissures	
2–3 toe syndactyly + single palmar crease	+	+	+	+			+			+		
Cardiac anomalies	+	+	+	+								
Ectodermal abnormalities	+	+		+			+	+	+			
Ocular abnormalities (strabismus; hyperopia)	+		+		+		+	+				
Genitourinary anomalies	+	+			+	+						
Brain abnormalities	+			+	+	+	+	+				
Hypotonia		+			+		+					
Other Features	Hearing loss				Ataxia, dysarthria				ASD	Myopia, microcornea	Tall stature	

“+”: present; empty cells: absent or data not available; DD: developmental delay; ID: intellectual disability; ASD: autism spectrum disorder.

**Table 3 genes-13-00780-t003:** Clinical data of 12q21 deleted patients from the DECIPHER database.

Patients	261,816	1582	1875	331,592	291,358	306,170	323,949	273,686
Growth retardation					+			
DD/ID	+	+	+	+		+	+	
Prominent forehead	+	+			+			
Hypertelorism								
Low set ears	+	+			+			
Short nose								
Other dysmorphisms		Upslanted palpebral fissure, wide mouth	Macrocephaly, brachycephaly		Long philtrum, relative macrocephaly, short neck, upslanted palpebral fissures			Highly arched eyebrow, upslanted palpebral fissures
2–3 toe syndactyly + single palmar crease					+			
Cardiac anomalies	+							
Ectodermal abnormalities		+	+					
Ocular abnormalities (strabismus; hyperopia)			+					
Genitourinary anomalies								
Brain abnormalities								
Hypotonia							+	
Other features		Joint laxity				Aggressive behavior, Peters anomaly		Corneal dystrophy, hand polydactyly, hypospadias

“+”: present; empty cells: absent or data not available; DD: developmental delay; ID: intellectual disability.

## Data Availability

Not applicable.

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
