# Peer review of "12q21 Interstitial Deletions: Seven New Syndromic Cases Detected by Array-CGH and Review of the Literature"

_genes, 2022, doi:10.3390/genes13050780_

Round 1

Reviewer 1 Report

This is a well-written manuscript with brief but informative introduction, appropriate methodology and very clear illustration of results both through tables and figures. The discussion was well organized and provided sound and logical conclusions as well. I have no major remark on this study. 

This study further examines the clinical and cytogenetic features of 12q21 DS, attempts to delineate the phenotype specific critical regions as identified in their own study patients (total number 7) and compares their findings with previously published reports and patients indexed in the DECIPHER database. This research question is highly clinically relevant and adds to the information available about the phenotype-genotype correlation in 12q21 deletion syndrome. It also helps to understand the roles and the effect of haploinsufficiency of different genes located within the deleted regions and fine tune the critical region and the contribution of genes in other regions for the phenotype.
This study builds up and strengthens the already available information through previous studies and ads up suggested gene attributions to specific phenotypes.
The methodology was appropriate to address the question above and that the data were well represented in the tables and figures. The figure clearly illustrated the genes in the deleted areas between different groups in the comparison. I think the discussion was clear and organized and the conclusions were consistent with the study findings and highlighted important remarks including what is known about the function of the genes in the deleted regions, the phenotypes that were found to be more specific to these regions and the importance of CVN analysis in syndromic patients in general. The references were appropriate as well.

Author Response

We thank the reviewer for the positive comments.

A revision of the English language has been performed as suggested.

Reviewer 2 Report

Summary

This manuscript presents the coordinates and phenotypes of seven (7) patients with 12q21 deletions that have not been previously reported in the medical literature. The authors also provide a review of previously published cases and DECIPHER database patients with deletions overlapping the 12q15q22 region. Patients 1-3 from this case series support the hypothesis of Niclass et al that a critical region at 12q21.2 may be responsible for some of the neurodevelopmental issue and dysmorphisms observed in patients with deletions involving the 12q21 region. Patients 4-7 provide evidence for another possible critical region at 12q21.31q21.33, similar to that presented by Akilapa et al, associated with neurodevelopmental phenotypes and a unique facial dysmorphism. These additional seven patients help to further fine-map the phenotypic associations for deletions in the 12q21 chromosomal region.

General Comments

The case series is well presented with significant phenotypic detail of the patient cohort (including presence and absence of recurrent phenotypes previously identified in patients with 12q21 deletion). Only one of the seven patients had whole exome sequencing completed, which did identify a possible alternate or concomitant diagnosis. The lack of whole-exome sequencing for the other six patients, although likely consistent with the standard of care for clinical management of patients in many jurisdictions, does raise concerns of missing alternate or concomitant diagnoses, and/or the inability to assess for “second hits” in genes within the deleted region that may unmask autosomal recessive phenotypes. This incomplete information could result in attribution of phenotypes observed in these patients to haploinsufficiency when an alternate mechanism may be at play. If possible, exome sequencing of all patients and interpretation of the copy number and sequence level data together would be preferred.

A thorough review of the 12q21 deletion literature is provided. Although small deletions in a few patients continue to refine the list of candidate genes, the genotype-phenotype correlations for specific sub-bands and/or genes remains a challenge. This report of seven additional patients, some with smaller deletions (though none were isolated to single genes), aids in further refinement of these correlations.

Specific comments

Line 97-101: Indicate the library preparation (exome baits) used for WES.  

Line 124: Readability, Should be “respectively”

Line 128: Readability, “20 up the 60 coding genes”.

Line 151: I suggest “The critical regions (CR) described in the literature are indicated with blue bars”. I suggest defining “PACD”, as figures should be able to stand alone without the text of the article. Is there a reference for PACD?

Line 165-168: Readability. Perhaps change “underlie” to “support”?

Line 172: Change to “As mentioned above”. Change “numbering” to “with”?

Line 175-180: SYT1 gene is predicted to be haploinsufficient based on HI% score, but Backer-Gordon syndrome may not necessarily act via a mechanism of haploinsufficiency. Reported variants in this disorder are missense variants that for the most part do not alter protein level or localization, but have been shown to cause variable slowing of the endocytic or exocytic rate following action potential stimulation (Baker et al 2018).

Line 181-182: Was the submitter to the DECIPHER database for Patient 331592 contacted to determine if additional phenotypes were present? Consider adding as “personal communication”.

Line 214: Consider providing coordinates for the 4.7 Mb region mentioned.

Line 229: The proposal of ALX1 as a plausible candidate for facial dysmorphisms observed in 12q21.31q21.32 deletion carriers, is not consistent with the autosomal recessive inheritance associated with frontonasal dysplasia 3 (FND3). Of the reported patients with homozygous 3.7Mb deletion that includes AXL1 (Uz et al, 2010) and splice site variants (Uz et al 2010 and Ullah et al 2017), the parents were heterozygous carriers and were reportedly unaffected.

Line 241: Is the sequence variant identified in Patient 5 classified as VUS or pathogenic? Is it a known loss of function variant?

Line 373: Reference 20 is incomplete.

Figure 1.

  1. Font size for Schluth et al 2008 is larger than the rest.
  2. “Akilopa_et_al_2015”, should read Akilapa

Figure 2.

  1. In Figure 2B was a control probe used in addition of the RP11-148D15 to show both the normal and deleted 12? As currently presented this figure does not add to the paper.
  2. Label 2B so that it is evident even when printed in black and white. Use of arrows or written label for the normal and deleted chromosomes 12 may help. Suggest making red arrow a white arrow so it will be visible when printed in black and white.
  3. Figure 2C is missing the chromosome ideogram.
  4. Figure 2E shows some coordinates, but they are too small to read and the genome build is not provided. Suggest removing these from figure 2E to be consistent with Figures 2A, 2C and 2D.

Table 1.

1.Provide genome build so that co-ordinates can be used to look up each deletion.

Table 2.

  1. Incorrect title. Switch with title from Table 3.
  2. Consider including the coordinates with appropriate genome build

Table 3.

  1. Incorrect title. Switch with title from Table 2.
  2. Consider including the coordinates with appropriate genome build
  3. “Oliverira” is misspelled.

Ethics statement: adequate

Data availability statement: Authors could consider inclusion of these variants in the DECIPHER and/or ClinVar databases if proper patient and/or parental consent is obtained.

Author Response

We thank the reviewer for the positive and constructive comments.

A revision of the English language has been performed as suggested.

Line 97-101: Indicate the library preparation (exome baits) used for WES.  

Enrichment kit added: SureSelect XT2 Clinical Research Exome (Agilent Technologies)

Line 124: Readability, Should be “respectively”

corrected

Line 128: Readability, “20 up the 60 coding genes”.

corrected

Line 151: I suggest “The critical regions (CR) described in the literature are indicated with blue bars”. I suggest defining “PACD”, as figures should be able to stand alone without the text of the article. Is there a reference for PACD?

Sentence corrected. PACD has been defined. Reference added

Line 165-168: Readability. Perhaps change “underlie” to “support”?

Changed

Line 172: Change to “As mentioned above”. Change “numbering” to “with”?

Changed

Line 181-182: Was the submitter to the DECIPHER database for Patient 331592 contacted to determine if additional phenotypes were present? Consider adding as “personal communication”. Not contacted

Line 214: Consider providing coordinates for the 4.7 Mb region mentioned.

Coordinates provided (78634011-83378230)

Line 241: Is the sequence variant identified in Patient 5 classified as VUS or pathogenic? Is it a known loss of function variant?

The variant has been classified as probably pathogenic (as reported), the variant in present in the HGMD database.

Line 373: Reference 20 is incomplete.

corrected

Figure 1.

corrected

Figure 2.

  1. In Figure 2B was a control probe used in addition of the RP11-148D15 to show both the normal and deleted 12? As currently presented this figure does not add to the paper. No control probe used
  2. Label 2B so that it is evident even when printed in black and white. Use of arrows or written label for the normal and deleted chromosomes 12 may help. Suggest making red arrow a white arrow so it will be visible when printed in black and white. White arrow added
  3. Figure 2C is missing the chromosome ideogram. Ideogram added
  4. Figure 2E shows some coordinates, but they are too small to read and the genome build is not provided. Suggest removing these from figure 2E to be consistent with Figures 2A, 2C and 2D. 2E modified

Table 1.

1.Provide genome build so that co-ordinates can be used to look up each deletion. hg19 added

Table 2.

Title corrected

Table 3.

Title corrected

Oliveira corrected